# Use of A MODIS Satellite-Based Aridity Index to Monitor Drought Conditions in Mongolia from 2001 to 2013

**Reiji Kimura [1,*] and Masao Moriyama [2]**

1   Arid Land Research Center, Tottori University, Tottori 680-0001, Japan
2   Graduate School of Engineering, Nagasaki University, Nagasaki 852-8521, Japan; matsu@nagasaki-u.ac.jp
*   Correspondence: rkimura@tottori-u.ac.jp; Tel.: +81-857-21-7031

**Abstract:** The 4D disasters (desertification, drought, dust, and *dzud*, a Mongolian term for severe winter weather) have recently been increasing in Mongolia, and their impacts on the livelihoods of humans has likewise increased. The combination of drought and *dzud* has caused the loss of livestock on which nomadic herdsmen depend for their well-being. Understanding the spatiotemporal patterns of drought and predicting drought conditions are important goals of scientific research in Mongolia. This study involved examining the trends of the normalized difference vegetation index (NDVI) and satellite-based aridity index (SbAI) to determine why the land surface of Mongolia has recently (2001–2013) become drier across a range of aridity indices (AIs). The main reasons were that the maximum NDVI ($NDVI_{max}$) was lower than the $NDVI_{max}$ typically found in other arid regions of the world, and the SbAI throughout the year was large (dry), although the SbAI in summer was comparatively small (wet). Under the current conditions, the capacity of the land surface to retain water throughout the year caused a large SbAI because rainfall in Mongolia is concentrated in the summer, and the conditions of grasslands reflect summer rainfall in addition to grazing pressure. We then proposed a method to monitor the land-surface dryness or drought using only satellite data. The correct identification of drought was higher for the SbAI. Drought is more strongly correlated with soil moisture anomalies, and thus the annual averaged SbAI might be appropriate for monitoring drought during seasons. Degraded land area, defined as annual $NDVI_{max} < 0.2$ and annual averaged SbAI > 0.025, has decreased. Degraded land area was large in the major drought years of Mongolia.

**Keywords:** aridity index; drought; land degradation; remote sensing; satellite-based aridity index

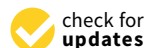

## 1. Introduction

In recent years, global warming has caused an increase in temperature and decrease in precipitation in drylands at high latitudes [1–4]. An increase in environmental stress associated with human activities concurrent with climate change may spread the damage caused by these three disasters [3,5].

In Mongolia, the damage caused by cold and snow is called "*dzud*" and is a natural disaster that can lead to significant livestock mortality and economic damage [6,7]. The authors in [7] have called desertification, drought, dust, and *dzud* the 4D-related hazards. The impact of *dzud* during the winter is strongly affected by the drought conditions (low pasture production) during the last summer. For example, *dzud* occurred from October 2009 to March 2010 due to the effect of drought during summer 2009 [7]. In Mongolia, about 30% of the workforce is engaged in raising livestock, and the 4D hazards, in addition to global warming, pose a serious threat to their livelihood. The authors in [8] have indicated that about 60% of the decline in vegetation in Mongolia from 1988 to 2008 can be attributed to decreases in rainfall and increases in temperature.

The Aridity Index (AI) is a useful metric of the dryness or wetness in arid regions. The AI is defined as the ratio of annual precipitation (Pr) to annual potential evaporation (Ep) and is a water-balance-based climatic index. The total area of arid regions, determined from

meteorological data collected from 1951 to 1980, was 41% of the terrestrial land surface, including Antarctica [9,10]. The corresponding percentage was 37% from 1981 to 2010 [11], and [12] have estimated that percentage to have been 39.5% from 2001 to 2013. These estimates suggest that the total area of arid regions has not changed or even decreased slightly since 1950.

The most widely used method to estimate aridity is the Palmer Drought Severity Index (PDSI) [13], not the AI. The PDSI is calculated using meteorological data and takes into consideration runoff, water supply, and the water retention capacity of the soil [14]. Some studies have estimated the PDSI in Mongolia [15–17], and all of them have concluded that the PDSI decreased (i.e., drought became more severe) from 2000 to 2010. However, there are some disadvantages to use of the PDSI. For example, the PDSI does not take into account changes in the spatial distributions of soil, vegetation, and hydrological processes [18].

With high resolution and frequency, satellite data offer advantages in monitoring environmental conditions in arid regions [19,20]. For example, the Moderate Resolution Imaging Spectroradiometer (MODIS) and Copernicus Missions (specifically Sentinel 1 or 2) have provided data since 2000 and 2014, respectively. Since lengthy cloudless periods are common in arid regions, much of the MODIS or Sentinel data are usable for global analyses.

Some drought indices are based on remote sensing [21,22]. Spectral reflectance has been widely used to calculate indices like the Normalized Difference Vegetation Index (NDVI) and normalized difference water index (NDWI) because the calculation procedures are simple [23]. The authors in [24] indicated that NDVI performed best in assessing land degradation compared with other indices using spectral reflectance like a soil-adjusted vegetation index (SAVI). Additionally, they revealed that thermal indices using land surface temperature (LST) were identified as the most influential variable for land degradation assessment. The authors in [25] have also suggested that a thermal index that uses the difference of the land surface temperature (LST) between day and night is much more useful as an indicator of water deficit. MODIS has provided daytime and nighttime LST data observed over equivalent locations every day, which have enabled the calculation of a thermal index since 2000.

The authors in [26] proposed a satellite-based aridity index (SbAI) that uses the difference of the LST between day and night, and the SbAI has already been validated and applied [12,26–29]. For example, years of major droughts in China have corresponded to years in which large increases in degraded land area were identified [28]. In addition, a comparison of the SbAI with the AI, that is, within Turc space (which is based on the water balance concept indicated by water limited to energy-limited lines) identified 15 categories in five zones: a stable zone, a zone transitioning toward dryness, a zone transitioning toward wetness, a dry zone, and a moist zone [30]. The authors of [30] have shown that the actual aridity was intensifying in most of Mongolia, though the climatic AI ranged from arid to semi-arid. The authors in [31] have demonstrated the NDVI relationship with precipitation and temperature in semi-arid regions and showed that the majority of sites displayed seasonal reversal, associated with transitions from water-limited to energy-limited conditions during wet winters.

Considering these past findings, the goal of this study was to examine the trends of NDVI and SbAI to determine why the land surface of Mongolia has recently become drier while the AI has ranged from arid to semi-arid and to propose a method to monitor the land-surface dryness or drought directly, using only satellite data.

## 2. Methods

### 2.1. Target Area and Analysis Period

Mongolia is a landlocked country surrounded by Russia, China, and Kazakhstan. The total land area is 1564,116 km$^2$. Mongolia is surrounded by high mountains and is located on a highland over 1500 m above sea level. The country has four distinct seasons, large temperature variations and little rainfall. The climate changes widely, not only due to

differences in altitude but also in latitude. The annual mean temperature is between −8 °C and 6 °C, and the annual mean precipitation is between 50 mm and 400 mm [6].

The AI, SbAI, and NDVI in Mongolia were calculated for latitudes of 41–53° N and longitudes of 87–120° E (Figure 1). With the exception of the woodlands in northern Mongolia, most of the land surface in Mongolia has been classified as typical grasslands and bare soil, including the Gobi Desert in southern Mongolia [32]. The red circles in Figure 1 indicate the SYNOP (surface synoptic observations) meteorological stations —Ulaanbaatar, Mandalgovi, and Tsogt-Ovoo from north to south—which are located in grassland, the boundary between grassland and bare soil, and bare soil, respectively.

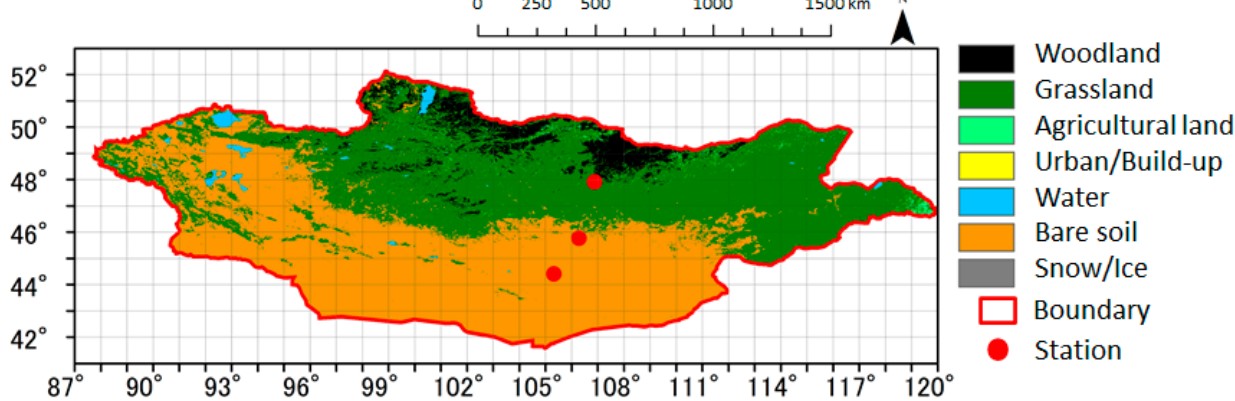

**Figure 1.** Land-use classification in Mongolia. Dots indicate the SYNOP meteorological stations (Ulaanbaatar, Mandalgovi, and Tsogt-Ovoo from north to south).

The time interval used to analyze the relationship between the AI and SbAI was 2001–2013. This period was chosen because precipitation data from the Global Precipitation Climatology Center's (GPCC) full data product (V7) are available throughout that time [12]. The GPCC has calculated precipitation for all global land areas during the target period through objective analysis of climatological average rainfall at the rain-gauge stations in its database.

The goal of this study is an examination of the trends of the NDVI and SbAI to determine why the land surface of Mongolia has recently (2001–2013) become drier. However, to establish the annual changes before and after 2001–2013, we calculated the NDVI from 1981 to 2000 using Advanced Very-High-Resolution Radiometer (AVHRR) data and from 2001 to 2020 using Moderate-Resolution Imaging Spectroradiometer (MODIS) data. We calculated the SbAI from 2001 to 2020 using MODIS data.

### 2.2. Data

The analysis of data regarding the relationship between the AI and SbAI from 2001 to 2013 was taken from [30], with a horizontal resolution of approximately 1° in both longitude and latitude.

We calculated the daily SbAI and NDVI from 2001 to 2020 using the Terra/MODIS data products MOD09CMG and MOD11C1 for surface reflectance and land surface temperature (LST) (https://modis-land.gsfc.nasa.gov/MODLAND_grid.html) (accessed on 27 May 2021). The spatial resolution of these two products was the same (0.05°). We used the "Collection 6" land product subsets web service to access and download the data [33].

We calculated daily NDVIs from 1981 to 2000 using the AVHRR surface reflectance (Channels 1 and 2) with 0.05° resolution (https://www.ncdc.noaa.gov/cdr/terrestrial/avhrr-surface-reflectance) (accessed on 27 May 2021). For long-term continuity of the NDVIs from the AVHRR and MODIS, we compared these two NDVIs in 2000 and obtained the following relationship with root mean squared errors (RMSE) = 0.09 (Figure 2):

$$\text{NDVI}_{\text{MODIS}} = 0.998 * \text{NDVI}_{\text{AVHRR}} + 1.975 * 10^{-5} \tag{1}$$

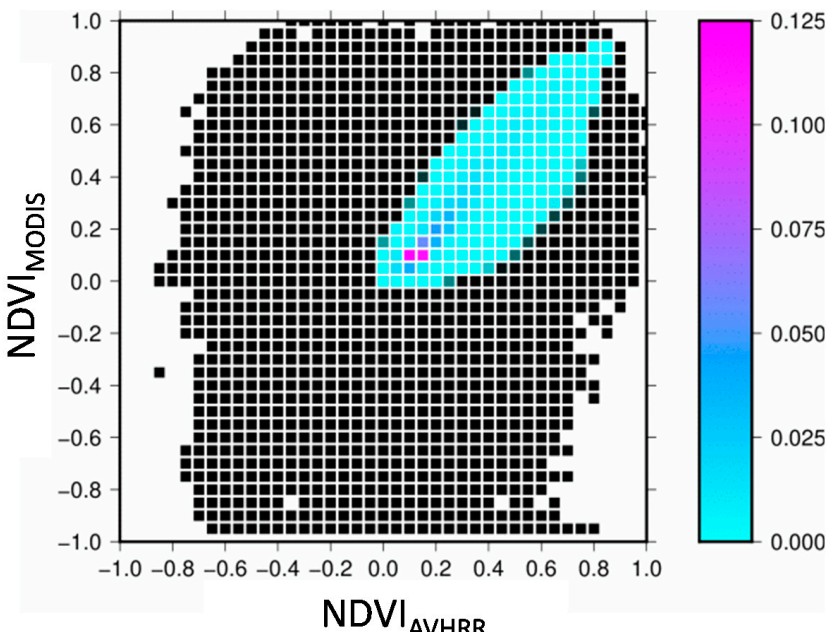

**Figure 2.** Relationship between the MODIS-NDVI and AVHRR-NDVI. Difference in color indicates the relative frequency.

In this study, Equation (1) was used to correct the NDVI$_{AVHRR}$ from 1981 to 2000.

A global land-cover map (GLCM) was used to characterize the distribution of land use in Mongolia (https://db.cger.nies.go.jp/dataset/landuse/en/) (accessed on 27 May 2021) (Figure 1). The GLCM is a raster image of Earth with a latitude-longitude resolution of 30 s, and it assigns the land cover of Earth into seven categories [34].

Annual rainfall data from 2001 to 2020 at Ulaanbaatar, Mandalgovi, and Tsogt-Ovoo (Figure 1) were downloaded from the Japan Meteorological Agency (JMA) website (http://www.data.jma.go.jp/gmd/cpd/monitor/climatview/frame.php) (accessed on 27 May 2021). When data were missing, the Climatic Resolution Unit Time Series monthly high-resolution gridded climate dataset was used to fill the gap (https://crudata.uea.ac.uk/cru/data/hrg/) (accessed on 27 May 2021).

### 2.3. Analytical Methods

The SbAI can be physically interpreted as a metric of the reciprocal heat capacity, which can be estimated from the ratio of the amplitude of the difference of the land surface temperature (LST) between day and night to the incident net solar radiation [26]. For a dry surface, the SbAI is large because the difference of the LST between day and night ($\Delta T_s$) is large. The $\Delta T_s$ is large because the low water content of the land surface causes its heat capacity to be low. For the analysis in this study, we used an annual average SbAI, which we calculated by averaging daily SbAIs, because the AI that we used to examine the relationship between the AI and SbAI was also an annual average [27].

In the analysis, we used the maximum NDVI in each year, because the amounts of vegetation were low, and information from NDVIs is lost in arid regions like Mongolia when annual averages are used [27,35]. In this study, the maximum NDVI occurred in August, and we used that NDVI as a metric of the potential for vegetation growth because the vegetation was strongly affected by the amount of precipitation prior to August (85% of annual rainfall in Mongolia occurs from April to July) [8,36–38].

The authors in [12,30] have used the SbAI to classify arid regions according to their actual degree of aridity (Table 1).

**Table 1.** Classification of arid regions using the SbAI and AI.

| Class | Range of SbAI | Range of AI |
|---|---|---|
| Hyper arid (HAr) | SbAI > 0.025 | AI < 0.05 |
| Arid (Ar) | $0.022 \leq SbAI \leq 0.025$ | $0.05 \leq AI < 0.2$ |
| Semi-arid (SAr) | $0.017 \leq SbAI < 0.022$ | $0.2 \leq AI < 0.5$ |
| Dry sub-humid (DSH) | $0.015 \leq SbAI < 0.017$ | $0.5 \leq AI < 0.65$ |

The range of the AI in each region, as defined by [9–11,39], is given in parentheses. The authors in [30] have subdivided these regions into 15 categories based on the SbAI and AI values of the four dryland regions (indicated by the double-headed red arrows along the *Y* and *X* axes) listed above (Figure 3). The stable zone (green), which includes points classified into the same dryland region by both the SbAI and AI, comprises categories 1, 2, 3, and 4. The transition zone, in which dryness is increasing (red), comprises categories 5 and 6, and the transition zone, in which wetness is increasing (blue), comprises categories 7, 8, and 9. In zones 10, 11, and 12, the magnitude of dryness is increasing in the dry zone (SbAI > 0.025), and zones 13, 14, and 15 are becoming wetter (SbAI < 0.015) (Figure 3).

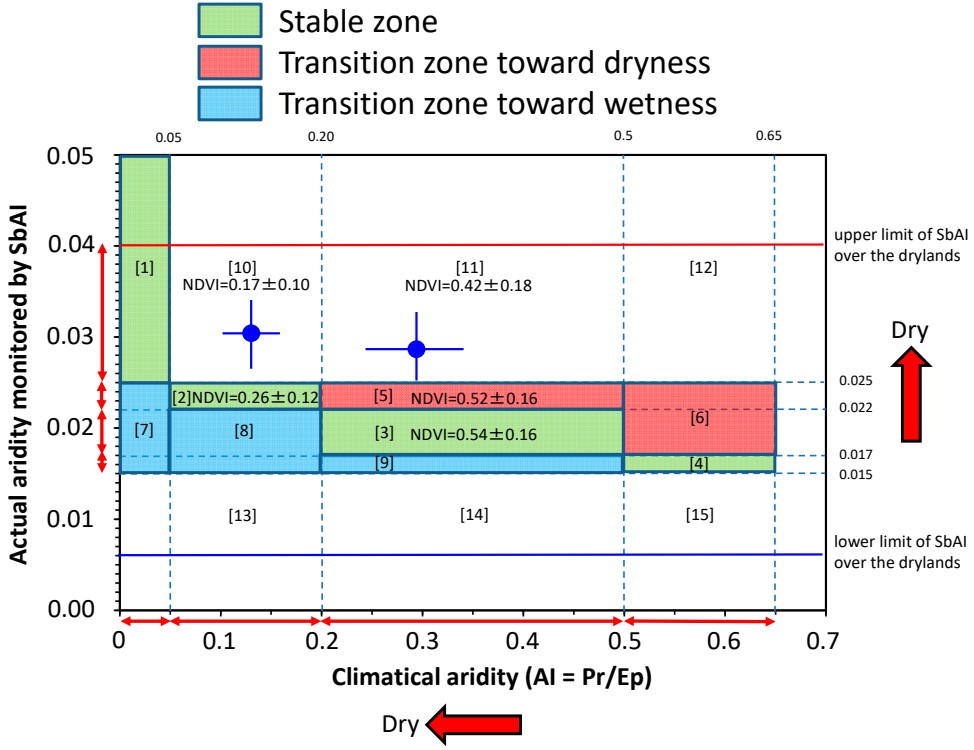

**Figure 3.** Relationship between the AI and SbAI, averaged over 2001–2013, and the 15 arid region categories. The red, double-headed arrows along the axes indicate the range of values of the indices that indicate hyper-arid, arid, semi-arid, and dry sub-humid regions. The blue dots with standard deviations indicate the actual ranges of the AIs and SbAIs in Mongolia (modified from [30]).

The authors in [27] examined the global distribution of annual maximum NDVI < 0.2 and annual averaged SbAI > 0.025 and defined areas that meet both these criteria as degraded land, which includes existing desert and land with both permanent and temporal dust erodibility. We examined yearly variation of degraded land area in Mongolia between 2001 and 2020 and discussed it in relation to drought.

## 3. Results

### 3.1. Distribution of Averaged AI in Mongolia from 2001 to 2013

The distribution of the averaged AI from 2001 to 2013 indicated that the northern region of Mongolia was dry sub-humid (DSH), the north central region was semi-arid (SAr), and the south region was arid (Ar) (Figure 4). A latitude of 47° N is the boundary between the SAr and Ar. The distribution of the AI corresponded to the distribution of land classification: northern Mongolia is woodland, north-central Mongolia is grassland, and southern Mongolia is bare soil (Figure 1).

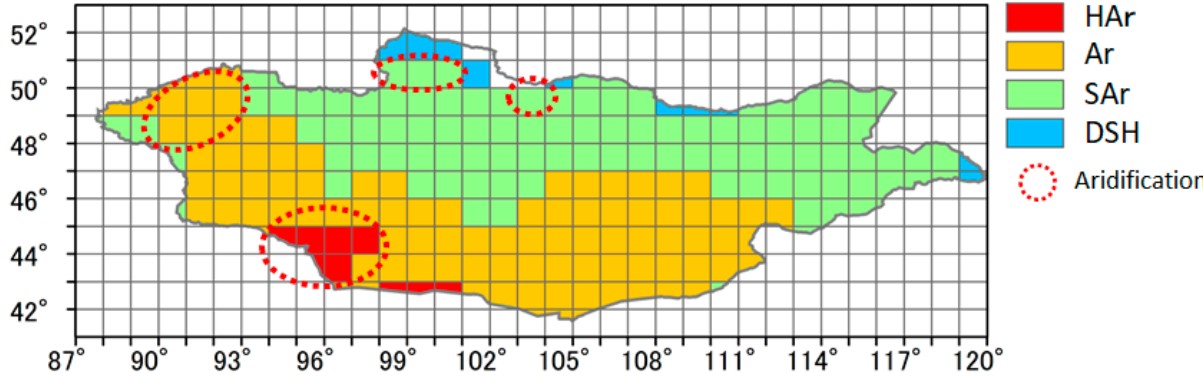

**Figure 4.** Spatial distribution of annual AI averaged during 2001–2013 in Mongolia with a resolution of 1° latitude × 1° longitude.

However, when the SbAIs were plotted against the AIs (blue dots with standard deviations in Figure 3), the areas that should have been in zones 2 and 3 were in zones 10 and 11 (Figure 5). This result indicates that the actual aridity in most of Mongolia was more severe than the climatic aridity. The authors in [40] have published a map of the distribution of aridity in Mongolia that shows a distribution of extremely strong to strong aridity, and middle to weak aridity similar to zones 10 and 11, respectively.

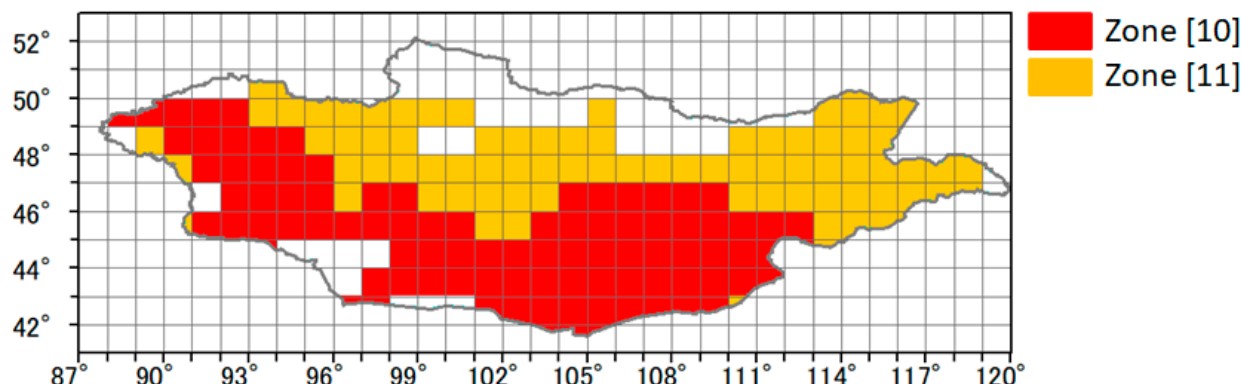

**Figure 5.** Spatial distribution of dry zone categories 10 and 11 during 2001–2013 in Mongolia with a resolution of 1° latitude × 1° longitude.

The authors in [30] have indicated that the yearly maximum of the NDVI (NDVI$_{max}$) in zones above the stable zone in Figure 3 has decreased, and the NDVI$_{max}$ in zones below the stable zone in Figure 3 has increased. The values of the NDVI$_{max}$ are therefore strongly related to the differences between the 15 zones in Figure 3. The values of the NDVI$_{max}$ in zones 10, 11, 2, 3, and 5 are shown in Figure 3, and the distribution of the annual NDVI$_{max}$, averaged over 2001 to 2013, is shown in Figure 6. The ranges of the NDVI$_{max}$ were 0.20 ± 0.11 and 0.51 ± 0.16 in zones 10 and 11, respectively (Figure 6), and were nearly consistent with the ranges of the NDVI$_{max}$ indicated in zones 10 and 5. That is, the smaller amount of vegetation may be one of the reasons why the land surface of Mongolia became drier from 2001 to 2013 across a range of AIs.

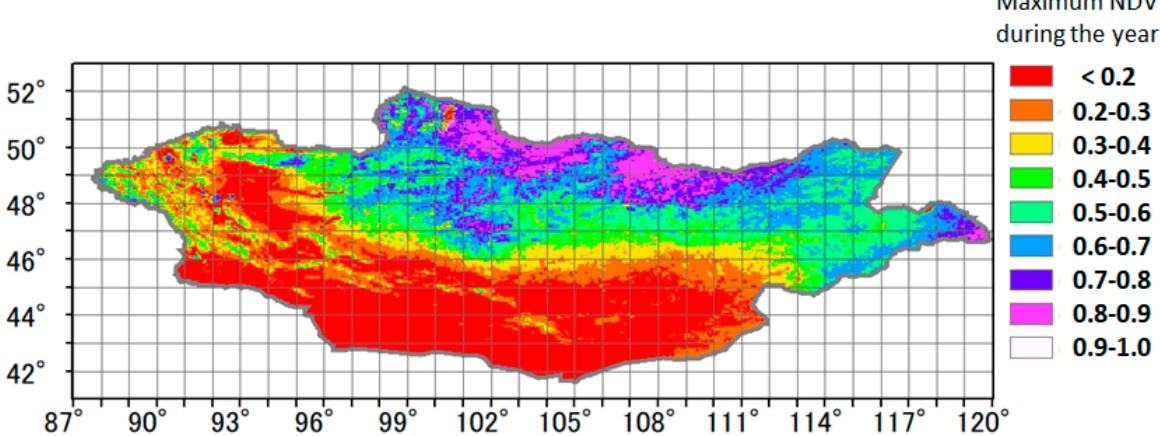

**Figure 6.** Spatial distribution of NDVI$_{max}$ values averaged during 2001–2013 in Mongolia with a resolution of 0.05° latitude × 0.05° longitude.

These results suggest the following characterizations of zones 10 and 11:

- Zone 10 is climatically an Ar region. From 2001 to 2013, however, this zone had less vegetation and was similar to an HAr region.
- Zone 11 is climatically a SAr region. The amount of vegetation in the summer is moderate. However, the actual water retention throughout the year in zone 11 inferred from its SbAI value was similar to that of an HAr region.

In Sections 3.2 and 3.3, we examine why the land surface of Mongolia became drier from 2001 to 2013 across a range of AIs by examining the trends of the NDVIs and SbAIs.

### 3.2. Difference of Climatic Conditions Using AI

We compared the AI distribution in Figure 4 to that from 1981–2010, calculated by [41], to identify the effects of climate change. Although a climatic trend toward aridity was apparent in some places (red circles), the distribution of AIs did not generally change (Figure 4).

Many studies have addressed trends of precipitation in Mongolia [7,8,17,42]. The authors in [43] examined the trend of annual precipitation from 1982 to 2010; they found that precipitation over Mongolia had been decreasing since 1993 (the trend was especially strong in northern and central Mongolia [17]) and the annual rainfall during 1994–2010 was about 30 mm lower than during 1982–1993. The authors in [7,8] found similar results. A decrease in annual rainfall by 30 mm has little effect on the classification of climates based on AI in Ar and SAr regions because the ranges of AIs in those categories are large: 0.05–0.2 and 0.2–0.5, respectively. The AI value itself may be reduced because of a decrease in rainfall and enhanced potential evaporation related to warmer temperatures [44]. The implication is that climatic effects are not revealed by a map of the distribution of AI values that reflect drier land surfaces in Mongolia because the range of AI values is large.

Monitoring the amount of vegetation will be an effective way to examine the effect of a decrease in rainfall by 30 mm over Mongolia. We therefore examined the trend of the NDVI$_{max}$ in August over Mongolia from 1981 to 2020 (Figure 7) because the NDVI$_{max}$ in August is sensitive to the amount of rainfall during the previous season [8,36,38]. In Mongolia, 85% of the annual rainfall is from April to July [37]. Figure 7 shows that the NDVI$_{max}$ decreased from 1994 to 2009. As previously mentioned, the annual rainfall during 1994–2010 decreased by about 30 mm compared with 1982–1993. The decreasing trend of the NDVI$_{max}$ up to 2009 was presumably due to decreased precipitation. However, the peak value of the NDVI$_{max}$ was only 0.39 in 1994, less than the average value of 0.4 in zones 2 and 3.

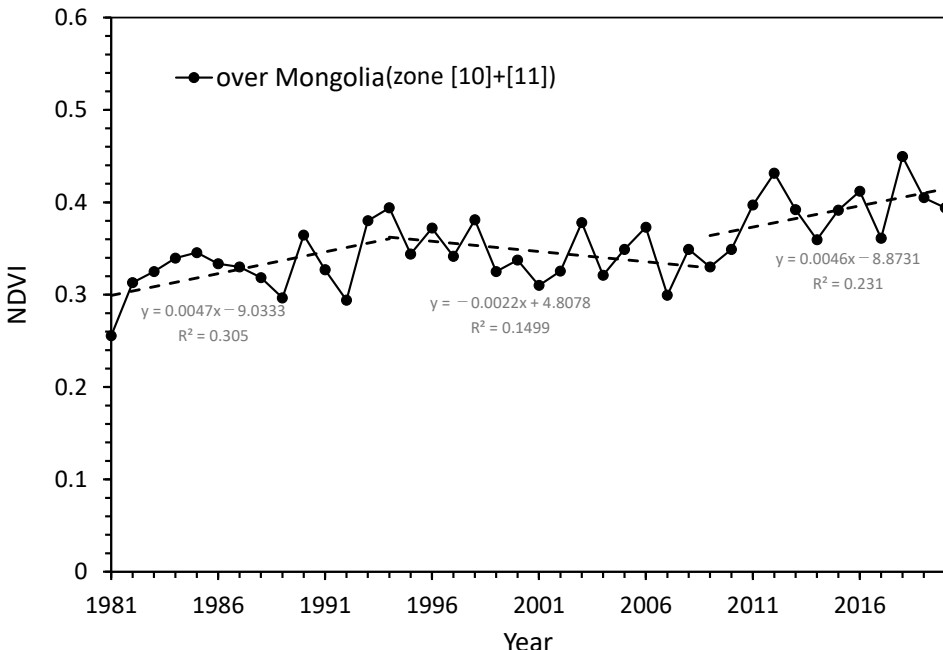

**Figure 7.** Annual changes (1981–2020) of the $NDVI_{max}$ in August over Mongolia. Dashed lines show the trends from 1981–1994, 1994–2009, and 2009–2020.

In contrast, an increasing trend of the $NDVI_{max}$ after 2009 can be found clearly. Since there have been few analyses of precipitation trends from 2001 to 2020 over Mongolia ([45] from 2000 to 2017; [23] from 2000 to 2016), we reexamined those trends at Ulaanbaatar, Mandalgovi, and Tsogt-Ovoo (Figure 8). At all three locations, annual rainfall increased after 2009, and those trends corresponded to an increase of the $NDVI_{max}$, which reached an averaged value of 0.4 in zones 2 and 3 (Figure 7). Based on an analysis of precipitation anomalies, the authors in [23] and [45] have also indicated that Mongolia became wetter from 2009 to 2017 compared with 2000–2008.

### 3.3. Trends of $NDVI_{max}$ and SbAI in Zones 10 and 11 during 2001–2020

We examined in detail the water retention of the land surface using the trends of the SbAI (averaged value in August and annual averaged value) and the $NDVI_{max}$ in August. In zone 10, the $NDVI_{max}$ decreased by a small amount from 2001 to 2009, and it increased very obviously after 2009 (Figure 9a). The average SbAI in August varied inversely with the $NDVI_{max}$, and the correlation between the two was high ($R^2 = 0.64$, $p < 0.001$). The averaged $NDVI_{max}$ from 2001 to 2010 was less than the limiting value of 0.2, below which land is considered to be degraded [27], but it increased after 2009. However, the $NDVI_{max}$ was smaller than the average value of 0.26 in Ar regions (zone 2). When the $NDVI_{max}$ exceeded 0.24, the averaged SbAI in August was lower than the limiting value of 0.025, below which land is considered to be degraded [27], but it was higher than 0.025 in many years. The annual averaged SbAI exceeded 0.03 in many years, and thus the environment of zone 10 appeared to be stressed in terms of water retention by the land surface through the year.

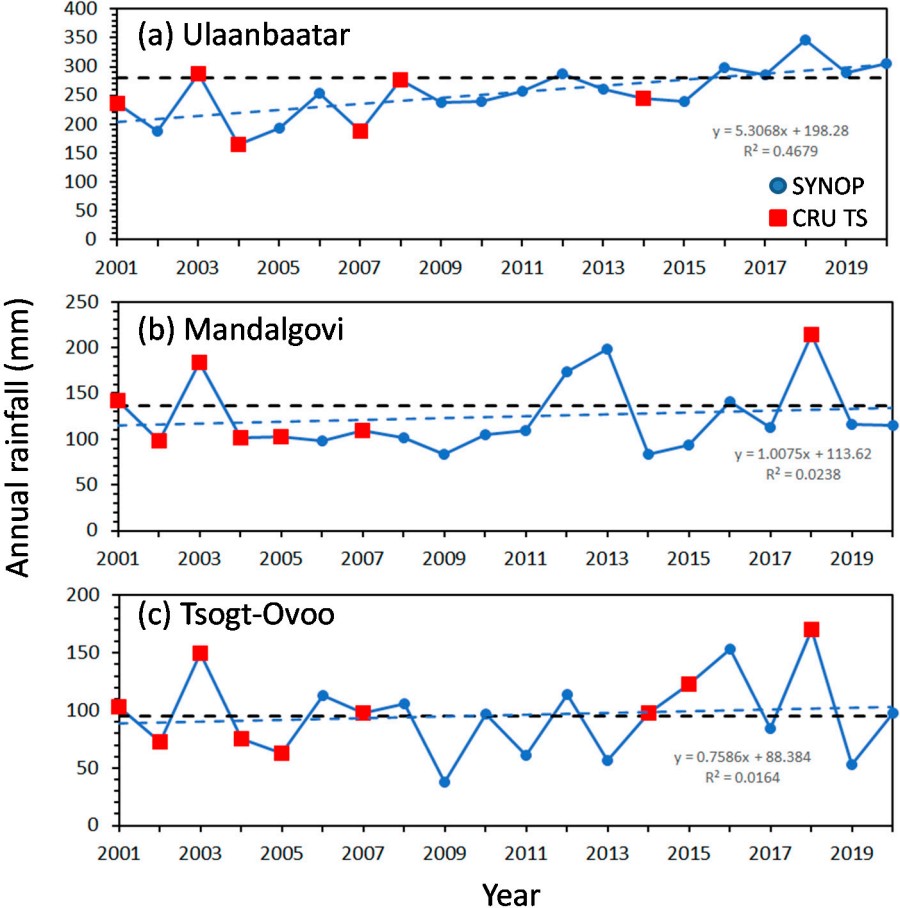

**Figure 8.** Annual changes (2001–2020) of annual rainfall in (**a**) Ulaanbaatar, (**b**) Mandalgovi, and (**c**) Tsogt-Ovoo. Black and blue dashed lines represent the normal values and trends during 1981–2010.

In zone 11, the $NDVI_{max}$ increased by a small amount from 2001 to 2009, and it increased very obviously after 2009 (Figure 9b). The averaged SbAI in August varied inversely with the $NDVI_{max}$, and the correlation between the two was high ($R^2 = 0.54$, $p < 0.001$). The $NDVI_{max}$ from 2001 to 2010 was lower than the general value of 0.54 in the SAr regions (zone 3), but it increased after 2009. The averaged SbAI in August from 2001 to 2009 was slightly lower than the limiting value of 0.025, below which land is considered to be degraded. However, the averaged SbAI in August after 2009 was substantially lower than 0.025, and it was even lower than 0.022, which is the upper bound for classification of a region as SAr (Figure 3). Although summers became wetter after 2009, water retention throughout the year was still low, because the annual averaged SbAI was 0.025–0.03.

The annual averaged SbAI was not necessarily correlated with the $NDVI_{max}$ ($R^2 = 0.31$, $p < 0.05$) (Figure 9b). For example, the $NDVI_{max}$ was lower in 2017 than in 2019, but the annual averaged SbAI was lower (wet) in 2017. There were similar relationships between the $NDVI_{max}$ and SbAI in 2010 and 2011. The authors in [45] indicated that water storage after summer in 2010 was higher than in 2011. It is inferred that precipitation in seasons other than summer affected water retention throughout the year. Numerical simulation results have shown that annual rainfall, especially rainfall during the winter, will increase over Mongolia from 2016 to 2035 and from 2081 to 2100 [37]. At the present time, it cannot definitively be concluded that the increasing trend of annual rainfall since 2009 (Figure 8) is in agreement with the simulation results. If the amount of precipitation increases enough that the annual averaged SbAI in zones 10 and 11 decreases to 0.025 and 0.022, respectively, the aridity in zones 10 and 11 will be close to climatically stable conditions.

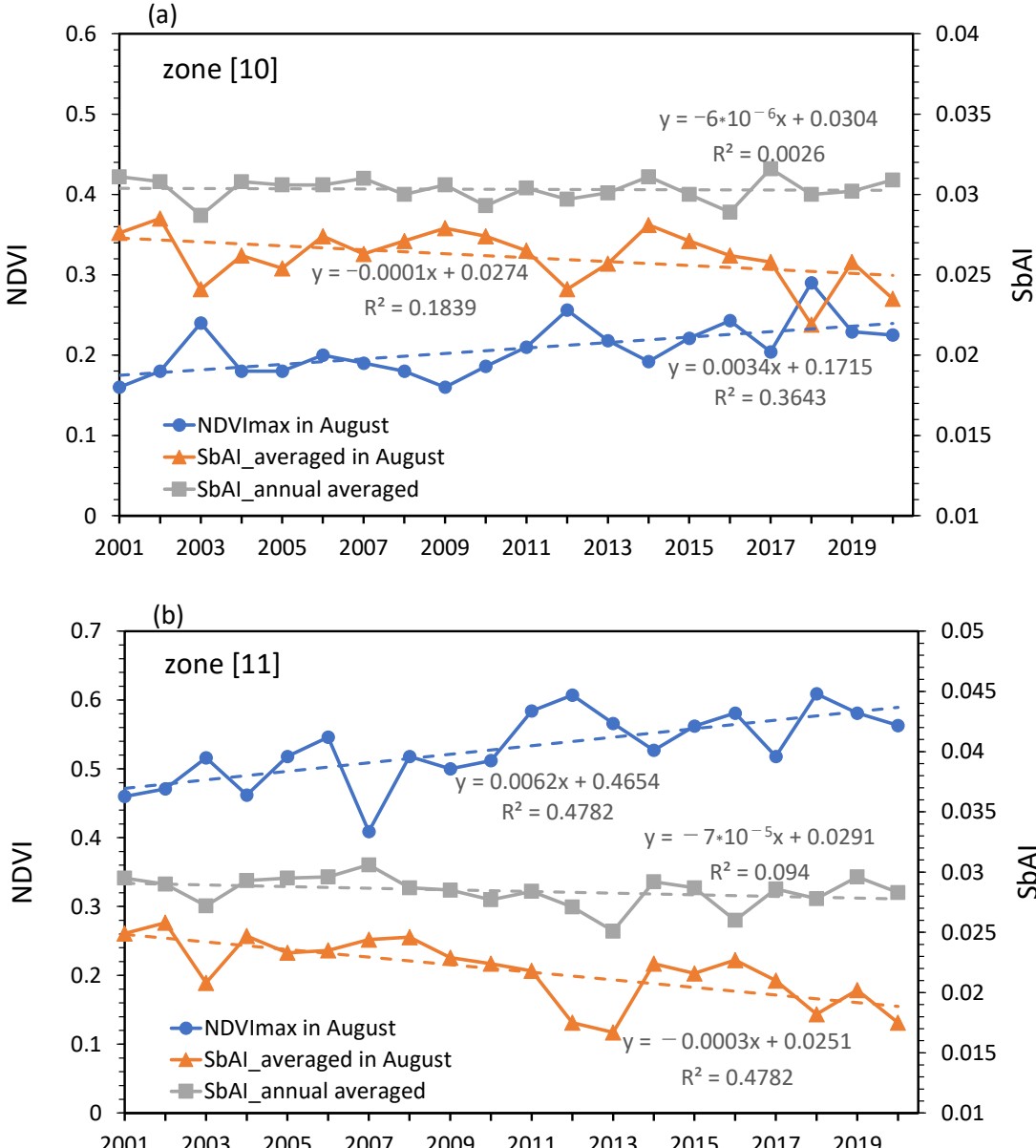

**Figure 9.** Annual changes (2001–2020) of the $NDVI_{max}$ in August, averaged SbAI in August, and annual averaged SbAI. (**a**) zone 10, (**b**) zone 11. Dashed lines show the trends of respective indices.

The annual averaged SbAI during the years shown in Figure 9b is the averaged value in zone 11, and thus there were wet regions with SbAIs below 0.022. For example, the distribution of annual averaged SbAIs in 2013 (the lowest SbAIs from 2001 to 2020) revealed wet regions with SbAIs below 0.022 (colored orange in Figure 10). The authors in [46] have indicated that although a large proportion of Mongolia's rangelands are not providing their potential ecosystem services, few have crossed an irreversible threshold of ecological change caused by current levels of grazing pressure. For the sustainable development of stock farming, continuous monitoring should be conducted to conserve the relatively wet regions (colored orange in Figure 10) and to prevent land degradation in nearly degraded regions (colored green in Figure 10).

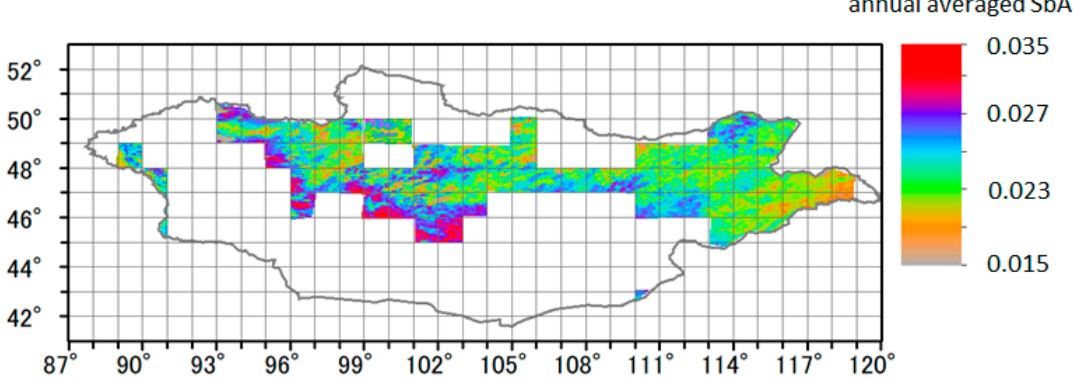

**Figure 10.** Spatial distribution of annual averaged SbAI values in zone 11 for 2003.

### 3.4. Detection of Drought Using SbAI

According to PDSI values, drought occurred frequently in all parts of Mongolia from 2000 to 2013 [7,16,37]. Since occurrences of *dzud* during the winter were strongly affected by drought conditions (low pasture production) during the preceding summer, understanding and predicting the characteristics of drought are of particular concern in Mongolia [15,37]. The authors in [15] have assessed drought frequency, duration, and severity over Mongolia from 2000 to 2010 using the PDSI and the standardized precipitation index (SPI). They have shown that droughts occurred in 2000, 2001, 2002, 2004, 2006, 2007, 2008, and 2009. Droughts therefore occurred in most years from 2000 to 2009 (red arrows in Figure 11). Figure 11 shows the yearly change in the $NDVI_{max}$, the averaged SbAI in August, and the annual averaged SbAI over Mongolia; the broken lines are the average values for the drought years. Both of the SbAI values equaled or exceeded these broken line averages during 2001–2009, but they have fallen below the broken lines in many years since 2009.

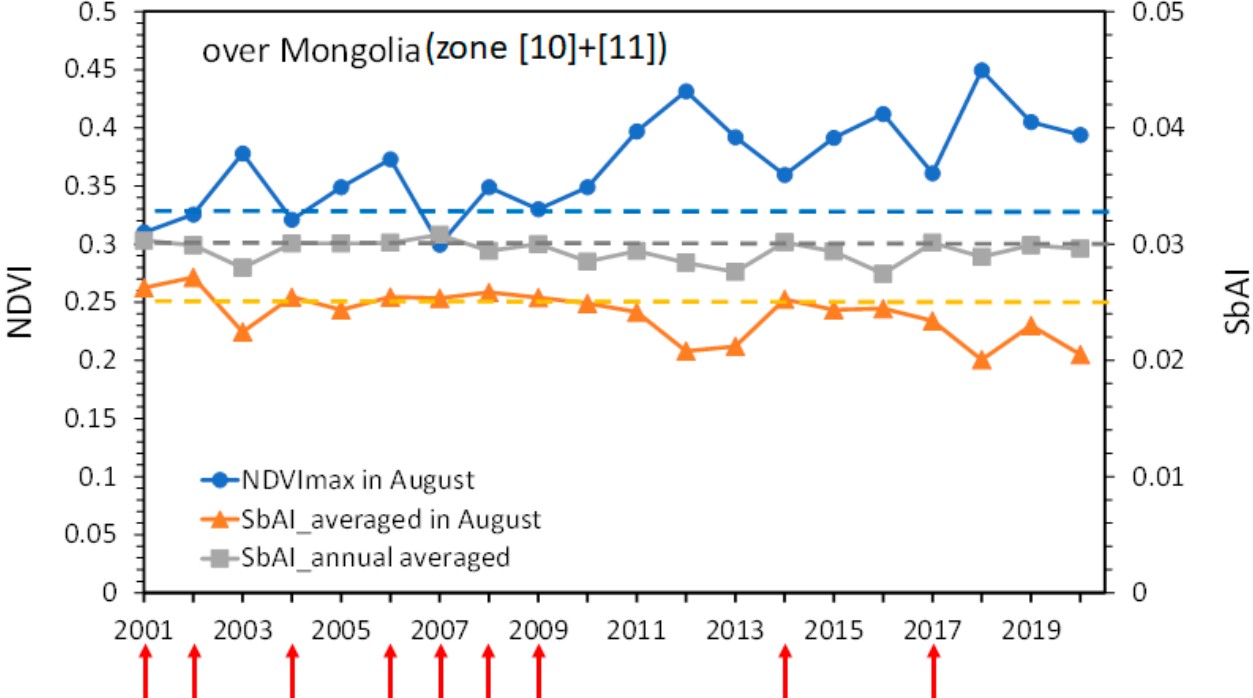

**Figure 11.** Annual changes (2001–2020) in the $NDVI_{max}$ in August, averaged SbAI in August, and annual averaged SbAI over Mongolia. Red arrows indicate drought years.

Drought years can be simply detected as follows:
$NDVI_{max} \leq 0.33$
averaged SbAI in August $\geq 0.025$
annual averaged SbAI $\geq 0.030$.

The correct identification of drought (presence or absence) was higher for the SbAI than for the NDVI during 2001–2009. In particular, the identification accuracy was 100% for the averaged SbAI in August. The primary reason for this accuracy is that the droughts during 2001–2010 were summer droughts that led to a reduction in water retention [15].

The years 2014 and 2017 have recently been drought years [28,38,45,47], and they could be detected by the annual averaged SbAI. Drought is more strongly correlated with soil moisture anomalies [36,48], and thus the annual averaged SbAI might be appropriate for monitoring drought during seasons other than summer.

Degraded land area, defined as annual $NDVI_{max} < 0.2$ and annual averaged SbAI > 0.025, has decreased ($R^2 = 0.24$, $p < 0.05$), especially since 2009 (Figure 12). Degraded land area was small in 2003, 2012, 2016, and 2018 but large in 2001, 2002, 2004, 2005, and 2009, which corresponded to the major drought years shown in Figure 11. Degraded land area can recover form one year to the next, as in 2017 to 2018. Since degraded land area was defined as areas including existing desert and land with both permanent and temporal dust erodibility [27], factors like an ecological processes and human impacts are also important in recovering degraded land [7].

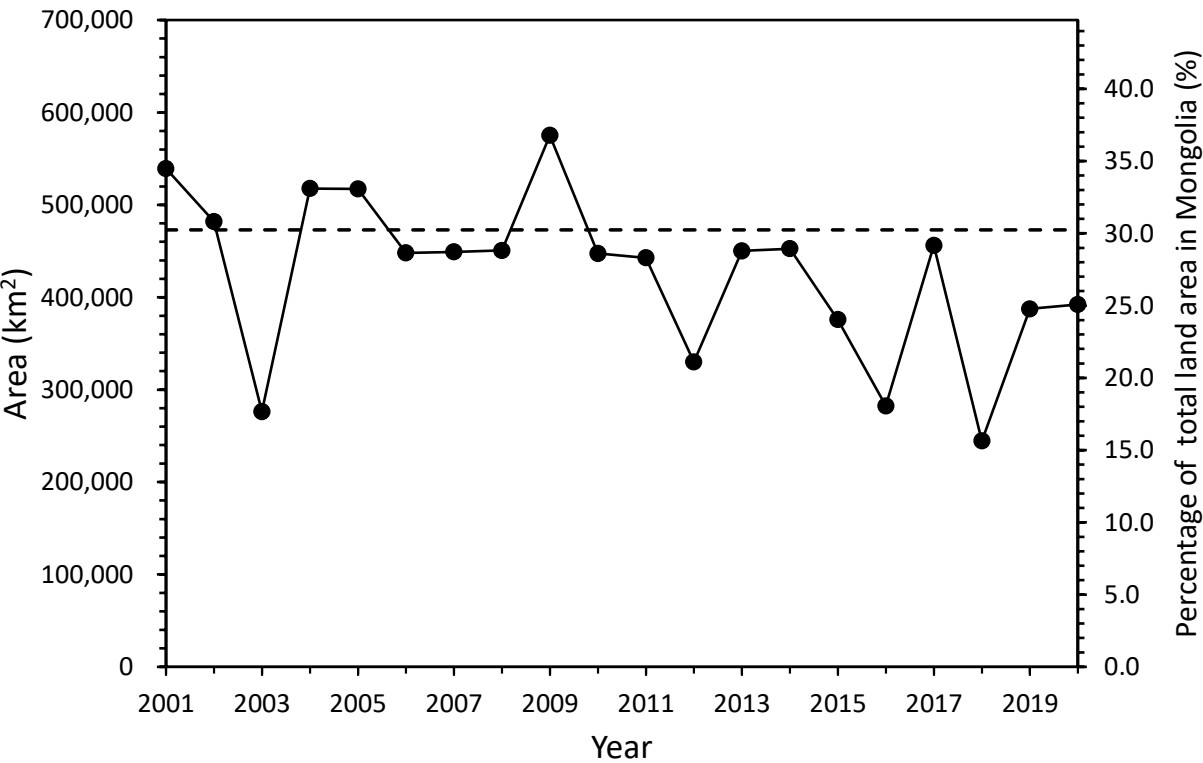

**Figure 12.** Annual changes of areas of degraded land and percentage of total land area in Mongolia. Dashed line represents the average extent of degraded land for 2001 to 2009.

The defined degraded land area should have been in zones 1 and 10 in Figure 3. Therefore, this method will be useful for general detection in very severe drought condition with a possibility of dust occurrences over Mongolia, particularly in zone 10 of Figure 5 (Figure 13). The spatial distribution of degraded land indicates that droughts have occurred frequently in southern-east Mongolia, that is, in Dundgovi, Omnogovi, and Dornogovi *aimags* (*aimag* is the first-level administrative subdivision). The authors in [49] also exhibited high risks of *dzud* in these provinces using social data from 1944 to 1993.

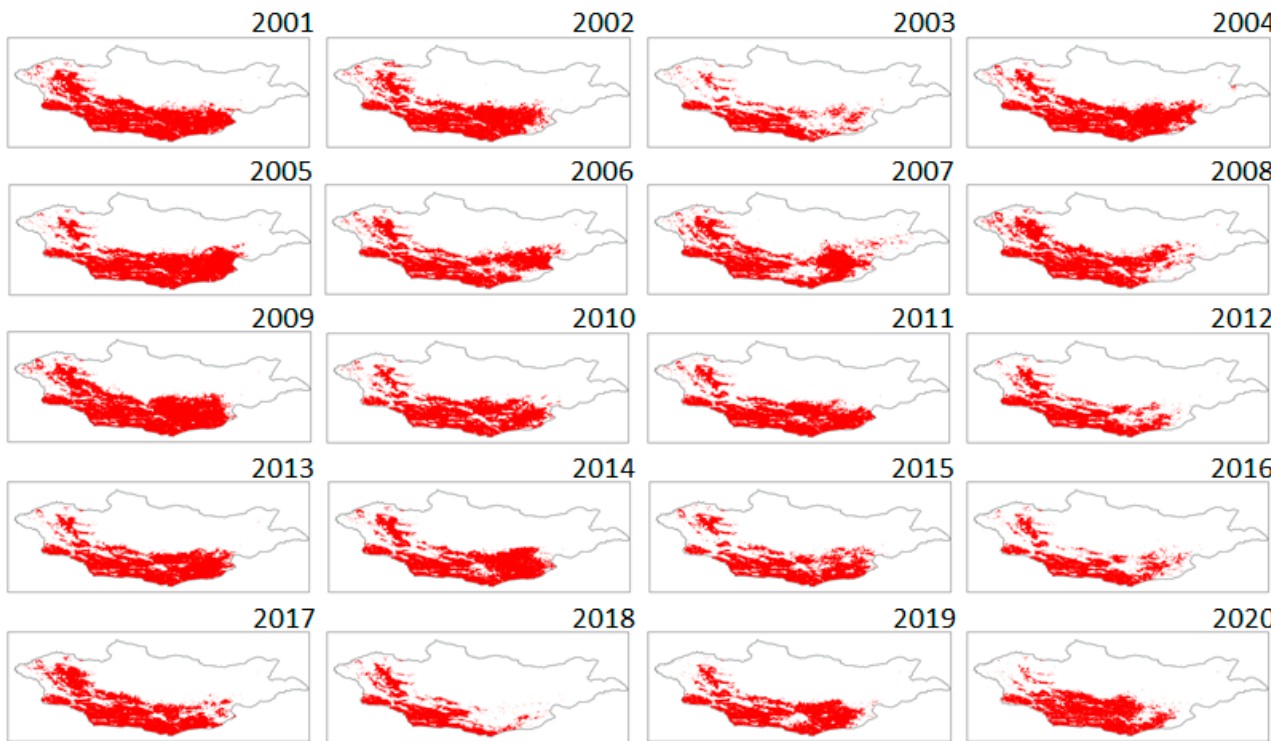

**Figure 13.** Spatial distribution of degraded land from 2001 to 2020.

## 4. Discussion

This study examined the reasons why the land surface in Mongolia has recently become drier across a range of AIs by examining the trends of the NDVI and SbAI. We then proposed a method to monitor drought conditions using only satellite data. Among explanations for why the SbAIs have been large within Mongolia are the following:

- The $NDVI_{max}$ was small compared with the $NDVI_{max}$ values in other Ar and SAr regions.
- Did Mongolia become drier climatically? Although the AI distribution was almost unchanged compared with 1981–2010, annual rainfall during 1994–2010 was about 30 mm less than during 1982–1993. There is a possibility that the amount of vegetation was sensitive to a rainfall decrease of 30 mm. In fact, the NDVImax had been decreasing up to 2010 after peaking in 1994. The NDVImax was small even at its peak value of 0.39 in 1994, and it did not reach its averaged value of 0.4 in zones 2 and 3.
- The SbAI during the summer was relatively small (wet). However, the SbAI through the year was large (dry). In Mongolia, most of the annual rainfall occurs from April to July, and that rainfall is reflected by the NDVImax in August. After August, vegetation is dried or eaten by livestock, and the land surface wetness decreases (large SbAI). At the same time, there is less rainfall during seasons other than summer.
- Under the current conditions, the capacity of the land surface to retain water leads to a large SbAI because the concentrated summer rainfall affects the growth of vegetation.

In contrast, the SbAI decreases when the annual rainfall and/or amount of rainfall increases during seasons other than summer. If the amount of precipitation, including precipitation during the winter, increases enough that the annual averaged SbAI decreases, the aridity of Mongolia will approach climatically stable conditions, and drought occurrences that are correlated with soil moisture anomalies will be less frequent. Since 2009, the $NDVI_{max}$ in August over Mongolia has tended to reach an average value of 0.4 in zones 2 and 3, and the frequency of drought years when SbAI values are over the threshold has also decreased.

For sustainable development in Mongolia, where 30% of the workforce is engaged in stock farming, continuous monitoring should be conducted to detect drought and prevent

land degradation. The remote sensing techniques proposed in this study, in addition to other drought indices that make use of meteorological or satellite data, will facilitate this monitoring. We hope that the usefulness of our method will be confirmed by other researchers and in other arid countries, and that our method will serve as the basis for an improved system based on remote sensing techniques that will promote sustainable development in arid regions throughout the world.

## 5. Conclusions

The purpose of this study was to examine the trends of the NDVI and SbAI to determine why the land surface of Mongolia has recently become drier; that is, when the SbAIs were plotted against the AIs, actual aridity in most of Mongolia was more severe than climatic aridity. The main reasons were that the $NDVI_{max}$ was lower than the $NDVI_{max}$ found in the other drylands of the world, and the SbAI throughout the year was large. Under the current conditions, the capacity of the land surface to retain water throughout the year caused the SbAI to be large because rainfall in Mongolia is concentrated in the summer, and the conditions of grasslands reflect summer rainfall.

A method was proposed to monitor land-surface dryness or drought using satellite data. The correct identification of drought was higher for the SbAI than for the NDVI. Drought is more strongly correlated with soil moisture anomalies, and thus the annual averaged SbAI might be appropriate for monitoring drought during seasons other than summer. Degraded land area, defined as annual NDVImax <0.2 and annual averaged SbAI > 0.025, has decreased. Degraded land area was small in 2003, 2012, 2016, and 2018 but large in 2001, 2002, 2004, 2005, and 2009, which corresponded to the major drought years in Mongolia. However, it must be noted that degradation is not caused by not only drought events but also ecological processes and grazing pressure in Mongolia [7].

**Author Contributions:** Conceptualization, R.K.; methodology, R.K.; software, M.M.; validation, R.K. and M.M.; formal analysis, R.K.; resources, M.M.; data curation, R.K. and M.M.; writing—original draft preparation, R.K.; writing—review and editing, R.K.; visualization, R.K.; supervision, R.K.; project administration, R.K.; funding acquisition, R.K. All authors have read and agreed to the published version of the manuscript.

**Funding:** This research was funded by Grant-in-Aid for Scientific Research, grant Number KAKENHI 19H04239.

**Institutional Review Board Statement:** Not applicable.

**Informed Consent Statement:** Not applicable.

**Acknowledgments:** We are very grateful to the reviewers who significantly contributed to the im-provement of this paper.

**Conflicts of Interest:** The authors declare that there are no conflict of interest.

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
