# Peer review of "Use of A MODIS Satellite-Based Aridity Index to Monitor Drought Conditions in Mongolia from 2001 to 2013"

_remotesensing, doi:10.3390/rs13132561_

Round 1
Reviewer 1 Report
Article Name: Monitoring of aridification in Mongolia with satellite-based indices
- I need to point out that the manuscript does snot have line numbers, which makes it more difficult to review.
Abstract:
- The first sentence in the abstract “Drylands are vulnerable to climate changes like global warming that can increase temperatures and decrease precipitation in arid regions of the Northern Hemisphere” should be revised, since the idea behind it is not clear.
- The portion of the abstract discussing “why” the region has become dryer, should be rewritten. As it reads, it seems like the authors suggests that the NDVI is the driver for the phenomena.
- Remarkable results should be enlisted in the abstract.
Introduction:
- This statement “In arid regions of the world, Desertification, Drought, and Dust are three natural disasters that begin with the letter “D”, might not be necessary, as an opening statement, one would expect a remark regarding the effects of these phenomena.
- “during the previous summer”à year?
- The study looks interesting, however, the authors need to discuss the “why”. As it is, by studying the current and past state of VIs and AIs the question seems more like a “How”. Are the authors discussing climatic trends? changes in land cover? If so, please add some of the discussion in the intro.
Methods:
- The study region needs to be describes better. How is it conformed? Climate? Vegetation types? Orography and relieve? Natural/political boundaries? (since it is fairly large, generalities like ecoregions or general features would work)
- Data acquisition and calibration seem proper.
- Discussion regarding these methods should be clarified according to the main question on the intro.
Results and Discussion:
- Desertification should be discussed, since it does not mean the same as aridity.
- “Monitoring the amount of vegetation will be an effective way to examine the effect of a decrease of rainfall by 30 mm”. I agree that analyzing vegetation trends would be an effective way for monitoring, however, was this decrement of 30 mm on PPT uniform over the entire territory?
- Figure 12: the author suggest that “degraded lands” can recover form one year to the next (ej. 2017-18 or 2015-16. The concept should be reviewed and adjusted.
The intro and abstract sections might need some english editing. The authors might want to reconsider claims regarding ecological processes for the entire region (ej. degradation, desertification, etc.), since these terms might be related to aridity, but are not the same concept. Overall an interesting approach for monitoring.
Author Response
RE: remotesensing-1256835
Title: Use of MODIS satellite-based aridity index to monitor drought conditions in Mongolia from 2001 to 2013
Authors: Reiji Kimura, Masao Moriyama
Response to referee comments
Enclosed please find the revised manuscript which we would like to resubmit for publication in Remote sensing. Revisions were made after carefully considering the comments provided by referees. Revisions are represented by yellow bold in the text.
- I need to point out that the manuscript does not have line numbers, which makes it more difficult to review.
(Answer)
We provided line number. I am very sorry for missing line numbers.
- Abstract: The first sentence in the abstract “Drylands are vulnerable to climate changes like global warming that can increase temperatures and decrease precipitation in arid regions of the Northern Hemisphere” should be revised, since the idea behind it is not clear. The portion of the abstract discussing “why” the region has become dryer, should be rewritten. As it reads, it seems like the authors suggests that the NDVI is the driver for the phenomena. Remarkable results should be enlisted in the abstract.
(Answer)
This sentence was deleted from the abstract. As you pointed, we changed and added the remarkable results to the abstract (Line 16-19, 23-27).
- Introduction: This statement “In arid regions of the world, Desertification, Drought, and Dust are three natural disasters that begin with the letter “D”, might not be necessary, as an opening statement, one would expect a remark regarding the effects of these phenomena. “during the previous summer” à year?
(Answer)
We deleted this sentence from introduction. And, we changed “previous summer” into “last summer”. (Line 40)
- The study looks interesting, however, the authors need to discuss the “why”. As it is, by studying the current and past state of VIs and AIs the question seems more like a “How”. Are the authors discussing climatic trends? changes in land cover? If so, please add some of the discussion in the intro.
(Answer)
We do appreciate for your important suggestion. We added the sentences into introduction with including past references. (Line 80-92).
- Methods: The study region needs to be describes better. How is it conformed? Climate? Vegetation types? Orography and relieve? Natural/political boundaries? (since it is fairly large, generalities like ecoregions or general features would work) Data acquisition and calibration seem proper. Discussion regarding these methods should be clarified according to the main question on the intro.
(Answer)
We added the information regarding boundary, topography, and climate briefly (Line 96-101).
Discussion regarding our proposed methods was added to the introduction (Line 60-76) .
- Results and Discussion: Desertification should be discussed, since it does not mean the same as aridity. “Monitoring the amount of vegetation will be an effective way to examine the effect of a decrease of rainfall by 30 mm”. I agree that analyzing vegetation trends would be an effective way for monitoring, however, was this decrement of 30 mm on PPT uniform over the entire territory?
(Answer)
As you pointed out, we changed “desertification” into “aridity” (Line 251-253). We added “over Mongolia” into the sentence as with Line 304. (Line 314)
- Figure 12: the author suggest that “degraded lands” can recover form one year to the next (ej. 2017-18 or 2015-16. The concept should be reviewed and adjusted.
(Answer)
We added following sentence; “Degraded land area can recover form one year to the next as 2017 to 2018. As discussed above, 2017 was drought year, and while 2018 was wet year owing to the heavy rainfall [38].” (Line 489-491)
- The intro and abstract sections might need some english editing. The authors might want to reconsider claims regarding ecological processes for the entire region (ej. degradation, desertification, etc.), since these terms might be related to aridity, but are not the same concept.
(Answer)
Finally, we are going to do English editing. We deleted the term “Desertification” from our results and discussion, and changed the title as “Use of MODIS satellite-based aridity index to monitor drought conditions in Mongolia from 2001 to 2013”.
Thank you for your comments and suggestion for our works. We added our appreciation to acknowledgements. (L. 587-588)
Reviewer 2 Report
I really found reading this manuscript interesting as authors have studied the monitoring of aridification in Mongolia using NDVI and SbAI. This Research is well organized, however there are certain things which must be done in order to make their research more insightful and strong. I would like to recommend authors – major revision, in order to improve the quality of the manuscript before considering it for publication.
The authors must provide line numbers to their article, as it becomes difficult to track the comments.
Comment -1: The title of the manuscript is perhaps general, and it should be more specific to better demonstrate the uniqueness of the indices used, as satellite indices give an impression that they have used several number of different indices.
Comment -2: Why MODIS is selected, please bring some light on other satellited products and compare its advantages and disadvantages. Also, I would recommend authors to add various studies such as Srivastava et al., 2017; Pettorelli et al., 2005 as they can increase the scientific weight and broader discussion into the introduction for using the MODIS product.
Srivastava, A., Sahoo, B., Raghuwanshi, N. S., & Singh, R. (2017). Evaluation of variable-infiltration capacity model and MODIS-terra satellite-derived grid-scale evapotranspiration estimates in a River Basin with Tropical Monsoon-Type climatology. Journal of Irrigation and Drainage Engineering, 143(8), 04017028.
Pettorelli, N., Vik, J. O., Mysterud, A., Gaillard, J. M., Tucker, C. J., & Stenseth, N. C. (2005). Using the satellite-derived NDVI to assess ecological responses to environmental change. Trends in ecology & evolution, 20(9), 503-510.
Comment -3: Why authors have focussed on NDVI, as there are vast amount of literature that utilises different indices such as SAVI, EVI, mNDWI etc. I would suggest to compare and contrast these indices atleast in the introduction, and they must highlight their use of NDVI.
Comment -4: It is seen that authors have not adequately focused on the literature review for relationship of NDVI with various variables like precipitation, aridity etc. I would strongly recommend the authors to add some recent studies that have applied the NDVI especially in arid and semiarid regions. Therefore I would recommend adding a very recent global study by Kumari et al., 2020 have shown the NDVI relationship with precipitation and temperature in semiarid regions. Elaborate more on how this paper differs or affirms the findings and conclusion of those studies. These points need to be clearly addressed in the introduction section. Hence, I would strongly recommend adding this study and comparing the results with their findings.
Kumari, N., Saco, P. M., Rodriguez, J. F., Johnstone, S. A., Srivastava, A., Chun, K. P., & Yetemen, O. (2020). The grass is not always greener on the other side: Seasonal reversal of vegetation greenness in aspect‐driven semiarid ecosystems. Geophysical Research Letters, 47(15), e2020GL088918. https://doi.org/10.1029/2020GL088918
Comment -5: Authors are recommended to clearly state the research gaps and their novelty of study in the Introduction.
Comment -6: I observed that authors have used datasets from different timeframes. For instance, they used AVHRR data from 1981-2000, then MODIS from 2001-2020. But in abstract they mention about only 2001-2013. I am not at all clear about this? Also, they used different timeframe for rainfall, SbAI in comparison to NDVI. They need to justify their dataset. Also, in order to avoid confusion, I recommend them to add a table for their datasets and the timeframe.
Comment -7: “SbAI to classify arid regions according to their actual degree of aridity as follows” – kindly provide in tabular form.
Comment -8: In Figure 1, I suggest to add north symbol and the scale to the figure too.
Comment -9: “For long-term continuity of the NDVIs from the AVHRR and MODIS, we compared these two NDVIs in 2000 and obtained the following relation”- The authors have shown the equation and RMS. First of all, they need to mention what is RMS. Second, they need to show this relationship either in scatter or linear plots, how well they were correlated.
Comment -10: In Figure 5, the NDVI legend need to be corrected- the first one can be <0.2, as “-0.2” looks like minus 0.2.
Comment -11: Figures 6-8, authors are just showing the time series plots, but they are required to show some trend analysis of how is it changing in such timescales.
Comment -12: Authors need to mention the date of access for all the datasets that they have used.
Author Response
RE: remotesensing-1256835
Title: Use of MODIS satellite-based aridity index to monitor drought conditions in Mongolia from 2001 to 2013
Authors: Reiji Kimura, Masao Moriyama
Dear Editor and Reviewers,
Enclosed please find the revised manuscript. Your consideration of this paper is greatly appreciated. If you have any questions, please contact me at
E-mail: [email protected]
Fax: +81-857-29-6199
Sincerely yours,
Reiji Kimura
Associate Professor
Arid Land Research Center
Tottori University
RE: remotesensing-1256835
Title: Use of MODIS satellite-based aridity index to monitor drought conditions in Mongolia from 2001 to 2013
Authors: Reiji Kimura, Masao Moriyama
Response to referee comments
Enclosed please find the revised manuscript which we would like to resubmit for publication in Remote sensing. Revisions were made after carefully considering the comments provided by referees. Revisions are represented by yellow bold in the text.
(Response to the comments by Reviewer 1)
- The authors must provide line numbers to their article, as it becomes difficult to track the comments.
(Answer)
We provided line number. I am sorry for missing line numbers.
- Comment -1: The title of the manuscript is perhaps general, and it should be more specific to better demonstrate the uniqueness of the indices used, as satellite indices give an impression that they have used several number of different indices.
(Answer)
We changed the title into “Use of MODIS satellite-based aridity index to monitor drought conditions in Mongolia from 2001 to 2013”.
- Comment -2: Why MODIS is selected, please bring some light on other satellited products and compare its advantages and disadvantages. Also, I would recommend authors to add various studies such as Srivastava et al., 2017; Pettorelli et al., 2005 as they can increase the scientific weight and broader discussion into the introduction for using the MODIS product.
(Answer)
We added following sentence with including your recommended papers; “With a high resolution and frequency, satellite data offer advantages in monitoring environmental conditions in arid regions [19,20]. For example, the Moderate Resolution Imaging Spectroradiometer (MODIS) and Copernicus Missions (specifically Sentinel 1 or 2) has provided data since 2000 and 2014, respectively. Because lengthy cloudless periods are common in arid regions, much of the MODIS or Sentinel data are usable for global analyses.” (Line 62 to 67, and references).
- Comment -3: Why authors have focussed on NDVI, as there are vast amount of literature that utilises different indices such as SAVI, EVI, mNDWI etc. I would suggest to compare and contrast these indices at least in the introduction, and they must highlight their use of NDVI.
(Answer)
We added following sentence with including recent papers by Bakhtiari et al. (2021);
“Some drought indices are based on remote sensing [21,22]. Spectral reflectance has been widely used to calculate indices like the NDVI and normalized difference water index (NDWI) because the calculation procedures are simple [23]. [24] indicated that NDVI performed best in assessing land degradation than other indices using spectral reflectance like a soil adjusted vegetation index (SAVI) in arid regions of Iran. And while, they revealed that thermal indices using land surface temperature (LST) was identified as the most influential variable for land degradation assessment. [25] have also suggested that a thermal index that uses the difference of the land surface temperature (LST) between day and night is much more useful as an indicator of water deficit. MODIS has provided daytime and nighttime LST data observed over equivalent locations every day that have enabled the calculation of a thermal index since 2000”. (Line 68 to 78, and references).
- Comment -4: It is seen that authors have not adequately focused on the literature review for relationship of NDVI with various variables like precipitation, aridity etc. I would strongly recommend the authors to add some recent studies that have applied the NDVI especially in arid and semiarid regions. Therefore I would recommend adding a very recent global study by Kumari et al., 2020 have shown the NDVI relationship with precipitation and temperature in semiarid regions. Elaborate more on how this paper differs or affirms the findings and conclusion of those studies. These points need to be clearly addressed in the introduction section. Hence, I would strongly recommend adding this study and comparing the results with their findings.
- Comment -5: Authors are recommended to clearly state the research gaps and their novelty of study in the Introduction.
(Answer)
We added following sentence with including your recommended paper; “In addition, a comparison of the SbAI with the AI, that is, within Turc space (which is based on the water balance concept indicated by water limited to energy limited lines) identified 15 categories in five zones: a stable zone; a zone transitioning toward dryness; a zone transitioning toward wetness; a dry zone; and a moist zone [30]. Results showed that the actual aridity was intensifying in most of Mongolia, though the climatic AI ranged from arid to semi-arid. [31] have shown the Normalized Difference Vegetation Index (NDVI) relationship with precipitation and temperature in semi-arid regions, and showed that majority of sites displayed seasonal reversal, associated with transitions from water-limited to energy-limited conditions during wet winters. With considering these past findings, the goal of this study was to examine the trends of the NDVI and SbAI to determine why the land surface of Mongolia has recently become drier while the AI has ranged from arid to semi-arid and to propose a method to monitor the land-surface dryness or drought directly using only satellite data.” (Line 82 to 94, and references)
- Comment -6: I observed that authors have used datasets from different timeframes. For instance, they used AVHRR data from 1981-2000, then MODIS from 2001-2020. But in abstract they mention about only 2001-2013. I am not at all clear about this? Also, they used different timeframe for rainfall, SbAI in comparison to NDVI. They need to justify their dataset. Also, in order to avoid confusion, I recommend them to add a table for their datasets and the timeframe.
(Answer)
We added following sentence in Methods; “The time interval used to analyze the relationship between the AI and SbAI was 2001–2013. This period was chosen because precipitation data from the Global Precipitation Climatology Center’s (GPCC) full data product (V7) are available throughout that time [12]. The GPCC has calculated precipitation for all global land areas during the target period by objective analysis of climatological average rainfall at the rain-gauge stations in its database. The goal of this study is an examination of the trends of the NDVI and SbAI to deter-mine why the land surface of Mongolia has recently (2001–2013) become drier. However, to know the annual changes before and after 2001-2013, we calculated the NDVI from 1981 to 2000 using Advanced Very-High-Resolution Radiometer (AVHRR) data and from 2001 to 2020 using Moderate-Resolution Imaging Spectroradiometer (MODIS) data. We calculated the SbAI from 2001 to 2020 using MODIS data.” (Line 114 to 125)
- Comment -7: “SbAI to classify arid regions according to their actual degree of aridity as follows” – kindly provide in tabular form.
(Answer)
We added Table 1 newly. (Page 5)
- Comment -8: In Figure 1, I suggest to add north symbol and the scale to the figure too.
(Answer)
We added north symbol and the scale into Figure 1.
- Comment -9: “For long-term continuity of the NDVIs from the AVHRR and MODIS, we compared these two NDVIs in 2000 and obtained the following relation”- The authors have shown the equation and RMS. First of all, they need to mention what is RMS. Second, they need to show this relationship either in scatter or linear plots, how well they were correlated.
(Answer)
We added Figure 2 newly (page 4), and explained RMSE (Line 139).
- Comment -10: In Figure 5, the NDVI legend need to be corrected- the first one can be <0.2, as “-0.2” looks like minus 0.2.
(Answer)
We corrected as you suggested in Figure 6. (Page 7)
- Comment -11: Figures 6-8, authors are just showing the time series plots, but they are required to show some trend analysis of how is it changing in such timescales.
(Answer)
In Figures 7 to 9, we showed trend analysis in the provided timescales.
- Comment -12: Authors need to mention the date of access for all the datasets that they have used.
(Answer)
We mentioned date of access for all the datasets. (Line 132, 137, 142, 147, and 150)
Thank you for your comments and suggestion for our works. We added our appreciation to acknowledgements. (L. 584-585)

Reviewer 3 Report
Comments and Suggestions for Authors:
The manuscript ‘remotesensing-1256835’ concerns satellite-based indices in arid region areas of the Northern Hemisphere. The research has focused on very important climate issues in arid regions in Mongolia. Nowadays, that devastating scenario we are seeing in most of the climate-affected countries. The author mentioned that the 4D disaster increased in Mongolia. To resolve the problem the scientific community needs to understand the drought pattern for making a prediction of drought conditions. The manuscript can be published in the Journal of Remote Sensing after major revisions.
Abstract: Should change and improve.
Introduction: What is ‘LST and MODIS’? Please explain clearly your methods, results, and conclusions.
Materials and Methods: Clear and understandable. However, you didn't mention anything about ‘Special distribution’, please explain this arrangement of phenomena here.
Results and discussion: You just mentioned which methods are good but you didn't classified, the aridification is most affecting in which land distribution areas. However, some questions have arisen from your investigations:
- Please check ‘page 5’ last three lines. You explained very well land distributions in your figure 1, can you explain please how your method can make a best examination for different land distribution areas.
- In addition, you would make a correlation between land distribution and their water retention capacity for different land distribution areas.
Conclusion:
Please develop the section based on your ‘results and discussion’.
Author Response
RE: remotesensing-1256835
Title: Use of MODIS satellite-based aridity index to monitor drought conditions in Mongolia from 2001 to 2013
Authors: Reiji Kimura, Masao Moriyama
Dear Editor and Reviewers,
Enclosed please find the revised manuscript. Your consideration of this paper is greatly appreciated. If you have any questions, please contact me at
E-mail: [email protected]
Fax: +81-857-29-6199
Sincerely yours,
Reiji Kimura
Associate Professor
Arid Land Research Center
Tottori University
RE: remotesensing-1256835
Title: Use of MODIS satellite-based aridity index to monitor drought conditions in Mongolia from 2001 to 2013
Authors: Reiji Kimura, Masao Moriyama
Response to referee comments
Enclosed please find the revised manuscript which we would like to resubmit for publication in Remote sensing. Revisions were made after carefully considering the comments provided by referees. Revisions are represented by yellow bold in the text.
(Response to the comments by Reviewer 2)
- Abstract: Should change and improve.
(Answer)
We changed the contents of abstract based on the results and discussion. (Line 9 to 27)
- Introduction: What is ‘LST and MODIS’? Please explain clearly your methods, results, and conclusions.
(Answer)
The meanings of LST and MODIS are explained (Line 63 and 73). In addition, we changed the contents of introduction as the reader can understand our methods and objectives (yellow bold in Introduction).
- Materials and Methods: Clear and understandable. However, you didn't mention anything about ‘Special distribution’, please explain this arrangement of phenomena here.
(Answer)
We are sorry that we cannot understand your comment here. So our answer may be incorrect. Maybe you pointed out the spatial distribution of land use in Figure 1. These distributions were generally made by the global land cover map (GLCM), not using our proposed methods. (Line 141 to 144)
- Results and discussion: You just mentioned which methods are good but you didn't classified, the aridification is most affecting in which land distribution areas. However, some questions have arisen from your investigations: Please check ‘page 5’ last three lines. You explained very well land distributions in your figure 1, can you explain please how your method can make a best examination for different land distribution areas. In addition, you would make a correlation between land distribution and their water retention capacity for different land distribution areas.
(Answer)
I do appreciate for your important comments. The goal of this study is an examination of the trends of the NDVI and SbAI to determine why the land surface of Mongolia has recently (2001–2013) become drier. Land use in Mongolia were classified into bare soil and grassland very simply (Fig. 1), and corresponded to zones 10 and 11, respectively. The effect of aridification on respective zones were described in line 425 to 434 and 505 to 511. Water retention capacity for different land distribution areas are difficult to examine in this paper. However, we have ever tried to estimate the water consumption in grasslands of north-east Asia (Kimura and Moriyama, 2020). Please let me consider about a correlation between land distribution and their water retention capacity for future developmental studies.
Kimura R, Moriyama M, 2020: Use of a satellite-based aridity index to monitor decreased soil water content and grass growth in grasslands of north-east Asia. Remote Sensing 12, 3556.
- Conclusion: Please develop the section based on your ‘results and discussion’.
(Answer)
We corrected “Conclusion” based on our results and discussion. (Line 561 to 576)
Thank you for your comments and suggestion for our works. We added our appreciation to acknowledgements. (L. 584-585)

Round 2
Reviewer 1 Report
Article Name: Monitoring of aridification in Mongolia with satellite-based indices
- I need to point out that the manuscript does snot have line numbers, which makes it more difficult to review.
Response to referee comments
Introduction: This statement “In arid regions of the world, Desertification, Drought, and Dust are three natural disasters that begin with the letter “D”, might not be necessary, as an opening statement, one would expect a remark regarding the effects of these phenomena. “during the previous summer” à year?
(Answer)
We deleted this sentence from introduction. And, we changed “previous summer” into “last summer”. (Line 40)
- Rev: The authors must point out which year it was. As it is, the time component is not defined.
The study looks interesting, however, the authors need to discuss the “why”. As it is, by studying the current and past state of VIs and AIs the question seems more like a “How”. Are the authors discussing climatic trends? changes in land cover? If so, please add some of the discussion in the intro.
(Answer)
We do appreciate for your important suggestion. We added the sentences into introduction with including past references. (Line 80-92).
- Rev: This section needs clarification; some grammatical errors can be seen in the new text. One of the sentences in the new text starts “Results showed…”, are the authors referring to previous studies? Other sentence starts with “have shown”… and needs to be completed.
Figure 12: the author suggest that “degraded lands” can recover form one year to the next (ej. 2017-18 or 2015-16. The concept should be reviewed and adjusted.
(Answer)
We added following sentence; “Degraded land area can recover form one year to the next as 2017 to 2018. As discussed above, 2017 was drought year, and while 2018 was wet year owing to the heavy rainfall [38].” (Line 489-491)
- There is a misconception here, the authors might want to review the concept of degradation (or the loss of function). It seems like the authors are mostly referring to changes in the mount of PPT.
- Rev: The following suggestion stands: The authors might want to reconsider claims regarding ecological processes for the entire region (ej. degradation, desertification, etc.), since these terms might be related to aridity, but are not the same concept. Overall an interesting approach for monitoring.
Author Response
RE: remotesensing-1256835
Title: Use of MODIS satellite-based aridity index to monitor drought conditions in Mongolia from 2001 to 2013
Authors: Reiji Kimura, Masao Moriyama
Response to referee comments
Enclosed please find the revised manuscript which we would like to resubmit for publication in Remote sensing. Revisions were made after carefully considering the comments provided by referees. Revisions are represented by yellow bold in the text.
(Response to the comments by Reviewer 1)
Rev: The authors must point out which year it was. As it is, the time component is not defined.
(Answer)
We added following sentence; “For example, dzud occurred during October 2009 to March 2010 due to the effect of drought during summer 2009 [7].” (Line 40-41)
Rev: This section needs clarification; some grammatical errors can be seen in the new text. One of the sentences in the new text starts “Results showed…”, are the authors referring to previous studies? Other sentence starts with “have shown”… and needs to be completed.
(Answer)
We integrated into “have shown”. (Line 86 to 87)
Rev: There is a misconception here, the authors might want to review the concept of degradation (or the loss of function). It seems like the authors are mostly referring to changes in the mount of PPT.
(Answer)
We are sorry for our misunderstanding. The definition of degradation in this study was discussed from line 233 to 236, and we added explanation of definition in degradation and factors (not only drought) affecting to degradation. (Line 489 to 493)
Rev: The following suggestion stands: The authors might want to reconsider claims regarding ecological processes for the entire region (ej. degradation, desertification, etc.), since these terms might be related to aridity, but are not the same concept. Overall an interesting approach for monitoring.
(Answer)
As you pointed out, we understand the differences among degradation, desertification, and aridity. Objective of this study was to monitor the land surface dryness or drought using satellite data, but degradation is not caused by not only the drought event but also the ecological process and grazing pressure. This important claim was added into conclusion. (Line 581-582)
I do appreciate for your careful suggestion and consideration for our works despite your crowded schedule.
sincerely yours,
Reiji Kimura
Arid Land Research Center, Tottori University
Japan
Reviewer 2 Report
The authors have addressed all the comments and in the meantime, they have improved the manuscript substantially. I recommend this manuscript be published in this journal.
Author Response
Dear Reviewer,
I do appreciate for your important comments and hospitality.
sincerely yours,
Reiji Kimura
Arid Land Research Center, Tottori University
Reviewer 3 Report
Thank you for your present manuscript structure that is more understandable than previous.
However, you need a minor revision to publish it.
Please Check: L16-L18, L529-L530, and L564-L565.
I think there are some words missing please check and fix them.
Author Response
RE: remotesensing-1256835
Title: Use of MODIS satellite-based aridity index to monitor drought conditions in Mongolia from 2001 to 2013
Authors: Reiji Kimura, Masao Moriyama
Response to referee comments
Enclosed please find the revised manuscript which we would like to resubmit for publication in Remote sensing. Revisions were made after carefully considering the comments provided by referees. Revisions are represented by yellow bold in the text.
-
Please Check: L16-L18, L529-L530, and L564-L565. I think there are some words missing please check and fix them.
(Answer)
Thank you for your careful reading and consideration. We changed properly about maximum NDVI. (Line 16 to 17, 533, 567)
Round 3
Reviewer 1 Report
I would prompt the authors to review the use of English, on some sections, and some minor writing errors. An example of the previous in in line 85, were “have” should start with Caps.
Author Response
Dear Reviewer 1,
Thank you for your comments to our works. We have changed as you suggested. Please follow the yellow bold line in the manuscript.
(Comments by Reviewer)
I would prompt the authors to review the use of English, on some sections, and some minor writing errors. An example of the previous in in line 85, were “have” should start with Caps.
(Answer)
We start with "The authors in" or "The author in" before the reference number.